# Face reconstruction from facial templates by learning latent space of a generator network

## Abstract

Face recognition systems are increasingly deployed in different applications. In these systems, a feature vector (also called facial embeddings or templates) is typically extracted from each face image and is stored in the system's database during the enrollment stage, which is later used for comparison during the recognition stage. In this paper, we focus on the template inversion attack against face recognition systems and propose a new method to reconstruct face images from facial templates. Within a generative adversarial network (GAN)-based framework, we learn a mapping from facial templates to the intermediate latent space of a pre-trained face generation network, from which we can generate high-resolution realistic reconstructed face images. We show that our proposed method can be applied in whitebox and blackbox attacks against face recognition systems. Furthermore, we evaluate the transferability of our attack when the adversary uses the reconstructed face image to impersonate the underlying subject in an attack against another face recognition system. Considering the adversary's knowledge and the target face recognition system, we define five different attacks and evaluate the vulnerability of state-of-the-art face recognition systems. Our experiments show that our proposed method achieves high success attack rates in whitebox and blackbox scenarios. Furthermore, the reconstructed face images are transferable and can be used to enter target face recognition systems with a different feature extractor model.

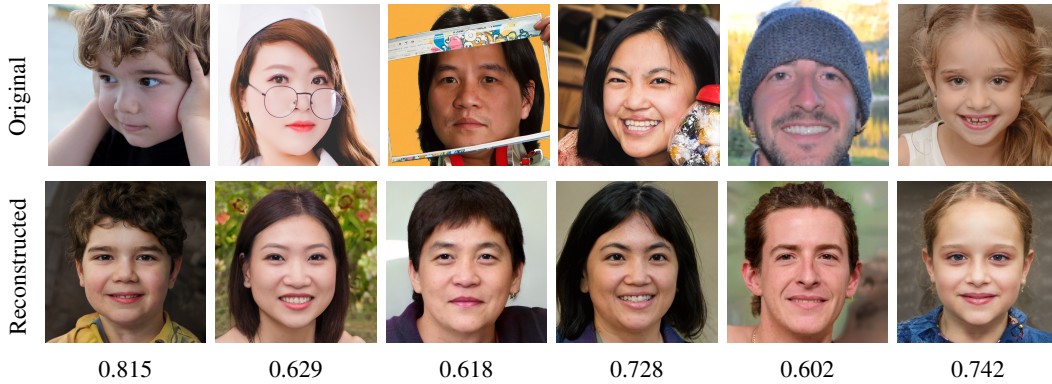

Figure 1: Sample face images from the FFHQ dataset and their corresponding reconstructed images using our template inversion method from ArcFace templates. The values below each image show the cosine similarity between the corresponding templates of original and reconstructed face images.

## 1 Introduction

Face recognition (FR) systems tend toward ubiquity, and their applications, which range from cell phone unlock to national identity system, border control, etc., are growing rapidly. Typically, in such systems, a feature vector (called embedding or template) is extracted from each face image using a deep neural network, and is stored in the system's database during the enrollment stage. During the

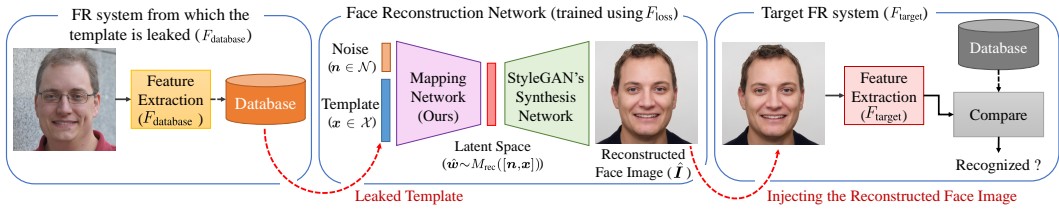

Figure 2: Block diagram of our proposed template inversion attack

recognition stage, either verification or identification, the extracted feature vector is compared with the ones in the system's database to measure the similarity of identities. Among potential attacks against FR systems (Galbally et al., 2014; Marcel et al., 2014; Biggio et al., 2015; Hadid et al., 2015; Mai et al., 2018), the template inversion (TI) attack significantly jeopardizes the users' privacy. In a TI attack, the adversary gains access to templates stored in the FR system's database and aims to reconstruct the underlying face image. Then, the adversary not only achieves privacy-sensitive information (such as gender, ethnicity, etc.) of enrolled users, but also can use reconstructed face images to impersonate.

In this paper, we focus on the TI attack against FR systems and propose a novel method to reconstruct face images from facial templates (Fig. 1 shows sample reconstructed face images using our proposed method). Within a generative adversarial network (GAN)-based framework, we learn a mapping from face templates to the intermediate latent space of StyleGAN3 (Karras et al., 2021), as a pre-trained face generation network. Then, using the synthesis part of StyleGAN3, we can generate high-resolution realistic face image. Our proposed method can be applied for *whitebox* and *blackbox* attacks against FR systems. In the *whitebox* scenario, the adversary knows the internal functioning of the feature extraction model and its parameters. However, in the *blackbox* scenario, the adversary does not know the internal functioning of the feature extraction model and can only use it to extract features from any arbitrary image. Instead, we assume that the adversary has a *whitebox* of another FR model, which can be used for training the face reconstruction network. We also evaluate the transferability of our attack by considering the case where the adversary uses the reconstructed face image to impersonate the underlying subject in an attack against another FR system (which has a different feature extraction model). Considering the adversary's knowledge and the target FR system, we define five different attacks, and evaluate the vulnerability of state-of-the-art (SOTA) FR systems. Fig. 2 illustrates the general black diagram of our proposed template inversion attack.

To elaborate on the contributions of our paper, we list them hereunder:

- We propose a novel method to generate high-resolution realistic face images from facial templates. Within a GAN-based framework, we learn the mapping from facial templates to the latent space of a pre-trained face generation network.

- We propose our method for *whitebox* and *blackbox* scenarios. While our method is based on the *whitebox* knowledge of the FR model, we extend our attack *blackbox* scenario, using another FR model that the adversary has access to.

- We define five different attacks against FR systems (based on the adversary's knowledge and the target system), and evaluate the vulnerability of SOTA FR models.

The remainder of the paper is organized as follows: Section 2 introduces the problem formulation and our proposed face reconstruction method. Section 3 covers the related works in the literature and compares them with our proposed method. Section 4 presents our experiential results. Finally, the paper is concluded in Section 5.

## 2 PROBLEM DEFINITION AND PROPOSED METHOD

In this paper, we consider a TI attack against a FR system based on the following threat model:

- *Adversary's goal*: The adversary aims to reconstruct a face image from a template, and use the reconstructed face image to enter the same or a different face recognition system, which we call the target FR system.
- *Adversary's knowledge:* The adversary knows a face template of a user enrolled in the FR system's database. The adversary also has either *whitebox* or *blackbox* knowledge of the feature extractor model in the same FR system.
- *Adversary's capability:* The adversary can present the reconstructed face image to the target FR system (e.g., using a printed photograph). However, for simplicity, we consider that adversary can inject the reconstructed face image as a query to the target FR system.
- *Adversary's strategy:* The adversary can train a face reconstruction model to invert facial templates and reconstruct underlying face images. Then, the adversary can use the reconstructed face images to inject as a query to the target FR system, to enter that system.

Let $F(.)$ denotes a facial feature extraction model, which gets the face image $I \in \mathcal{I}$ and extracts facial template $x = F(I) \in \mathcal{X}$. According to the threat model, the adversary has access to the target facial template $x_{\text{database}} = F_{\text{database}}(I)$ and aims to generate a reconstructed face image $\hat{I}$. Then, the adversary can use the reconstructed face image $\hat{I}$ to impersonate the corresponding subject and attack a target FR system with $F_{\text{target}}(.)$, which might be different from $F_{\text{database}}(.)$.

To train a face reconstruction model, we can use a dataset of face images $\{I_i\}_{i=1}^N$ with $N$ face images (no label is required), and generate a training dataset $\{(x_i, I_i)\}_{i=1}^N$, where $x_i = F_{\text{database}}(I_i)$. Then, a face reconstruction model $G(.)$ can be trained to reconstruct face image $\hat{I} = G(x)$ given each facial template $x \in \mathcal{X}$. To train such a face reconstruction model, we consider a multi-term face reconstruction loss function as follows:

$$\mathcal{L}_{\text{rec}} = \mathcal{L}_{\text{pixel}} + \mathcal{L}_{\text{ID}}, \tag{1}$$

where $\mathcal{L}_{\text{pixel}}$ and $\mathcal{L}_{\text{ID}}$ indicate pixel loss and ID loss, respectively, and are defined as:

$$\mathcal{L}_{\text{pixel}} = \mathbb{E}_{x \sim \mathcal{X}}[\|I - G(x)\|_2^2], \tag{2}$$

$$\mathcal{L}_{\text{ID}} = \mathbb{E}_{x \sim \mathcal{X}}[\|F_{\text{loss}}(I) - F_{\text{loss}}(G(x))\|_2^2]. \tag{3}$$

The pixel loss is used to minimize the pixel-level reconstruction error of the generated face image. The ID loss is also used to minimize the distance between facial templates extracted by $F_{\text{loss}}(.)$ from original and reconstructed face images. In Eq. 3, $F_{\text{loss}}(.)$ denotes a feature extraction model that the adversary is assumed to have complete knowledge of its parameters and internal functioning. Based on the adversary's knowledge of $F_{\text{database}}(.)$ (i.e., *whitebox* or *blackbox* scenarios), $F_{\text{loss}}(.)$ might be the same or different from $F_{\text{database}}(.)$.

For the face reconstruction model, we consider StyleGAN3 (Karras et al., 2021), as a pre-trained face generation network. The Style-GAN3 model is trained on a dataset of face images using a GAN-based framework that can generate high-resolution and realistic face images. The structure of StyleGAN3 is composed of two networks, mapping and synthesis networks. The mapping network $M_{\text{StyleGAN}}(.)$ gets a random noise $z \in \mathcal{Z}$ and generates an intermediate latent code $w = M_{\text{StyleGAN}}(z) \in \mathcal{W}$. Then, the latent code $w$ is given to the synthesis network $S_{\text{StyleGAN}}(.)$ to generate a face image. In our training process, we fix the synthetic

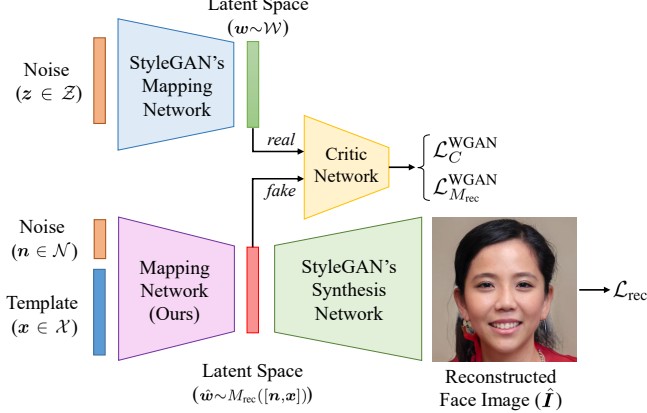

Figure 3: Block diagram of our face reconstruction network.

network $S_{\text{StyleGAN}}(.)$ and train a new mapping $M_{\text{rec}}(.)$ to generate $\hat{w}$ corresponding to the given facial template $x \in \mathcal{X}$. Then, the generated latent code $\hat{w}$ is given to the synthesis network $S_{\text{StyleGAN}}(.)$

to generate the reconstructed face image $\hat{I} = S_{\text{StyleGAN}}(\hat{w})$. We can train our new mapping $M_{\text{rec}}(.)$ using our reconstruction loss function as in Eq. 1. However, to obtain a realistic face image from the generated $\hat{w}$ through the pre-trained synthetic network $S_{\text{StyleGAN}}(.)$, the generated $\hat{w}$ needs to be in the distribution $\mathcal{W}$; otherwise, the output may not look like a real human face. Hence, to generate $\hat{w}$ vectors such that they have the same distribution as StyhleGAN's intermediate latent, $w \in \mathcal{W}$, we use a GAN-based framework to learn the distribution $\mathcal{W}$. To this end, we use the Wasserstein GAN (WGAN) Arjovsky et al. (2017) algorithm to train a critic network $C(.)$ which critics the generated $\hat{w}$ vectors compared to the real StyleGAN's $w \in \mathcal{W}$ vectors, and simultaneously we optimize our mapping network to generate $\hat{w}$ vectors with the same distribution as $\mathcal{W}$. Hence, we can consider our mapping network $M_{\text{rec}}(.)$ as a conditional generator in our WGAN framework, which generates $\hat{w} = M_{\text{rec}}([n, x])$ given a facial template $x \in \mathcal{X}$ and a random noise vector $n \in \mathcal{N}$. Then, we can train our mapping network and critic network using the following loss functions:

$$\mathcal{L}_C^{\text{WGAN}} = \mathbb{E}_{w \sim \mathcal{W}}[C(w)] - \mathbb{E}_{\hat{w} \sim M_{\text{rec}}([n, x])}[C(\hat{w})] \tag{4}$$

$$\mathcal{L}_{M_{\text{rec}}}^{\text{WGAN}} = \mathbb{E}_{\hat{w} \sim M_{\text{rec}}([n, x])}[C(\hat{w})] \tag{5}$$

In a nutshell, we train a new mapping network $M_{\text{rec}}(.)$ using our reconstruction loss function in Eq. 1, and also optimize $M_{\text{rec}}(.)$ within our WGAN framework using Eq. 5. Simultaneously, we also train the critic network $C(.)$ within our WGAN using Eq. 4 to learn the distribution of StyleGAN's intermediate latent space $\mathcal{W}$ and help our mapping network $M_{\text{rec}}(.)$ to generate vectors with the same distribution as $\mathcal{W}$. Fig. 3 depicts the block diagram of the proposed method. We should note that our mapping network $M_{\text{rec}}(.)$ has 2 fully connected layers with Leaky ReLU activation function.

In our problem formulation, we consider three different feature extraction models, including $F_{\text{database}}(.)$, $F_{\text{loss}}(.)$, and $F_{\text{target}}(.)$. Hence, based on the adversary's knowledge and the target system, we can consider five different attacks:

- **Attack 1:** The adversary has *whitebox* knowledge of the system from which the template is leaked and want to attack the same system (i.e., $F_{\text{database}} = F_{\text{loss}} = F_{\text{target}}$).

- **Attack 2:** The adversary has *whitebox* knowledge of the feature extractor of the system from which the template is leaked, but aims to attack to a different FR system (i.e., $F_{\text{database}} = F_{\text{loss}} \neq F_{\text{target}}$).

- **Attack 3:** The adversary wants to attack the same system from which the template is leaked, but has only *blackbox* access to the feature extractor of the system. Instead, we assume that the adversary has the *whitebox* knowledge of another FR model to use for training (i.e., $F_{\text{database}} = F_{\text{target}} \neq F_{\text{loss}}$).

- **Attack 4:** The adversary aims to attack a different FR system than the one from which the template is leaked. In addition, the adversary has *whitebox* knowledge of the feature extractor of the target system (i.e., $F_{\text{database}} \neq F_{\text{loss}} = F_{\text{target}}$).

- **Attack 5:** The adversary aims to attack a different FR system from which the template is leaked and has only *blackbox* knowledge of both the target system and the one from which the template is leaked. However, the adversary instead has the *whitebox* knowledge of another FR model to use for training (i.e., $F_{\text{database}} \neq F_{\text{loss}} \neq F_{\text{target}}$).

In the attack 1 and attack 2, the adversary has the *whitebox* knowledge of the system from which the template is leaked (i.e., $F_{\text{database}}(.)$) and uses the same model as $F_{\text{loss}}(.)$ for training the reconstruction network. However, in attacks 3-5, the adversary has the *blackbox* knowledge of the system from which the template is leaked, and therefore uses another FR model as $F_{\text{loss}}(.)$. Comparing the knowledge of the adversary in these attacks, we expect that attack 1 be the easiest attack for the adversary and attack 5 be the most difficult one.

## 3  RELATED WORKS

Table 1 compares our proposed method with related works in the literature. Generally, the methods for TI attack against FR systems, can be categorized based on different aspects, including the resolution of generated face images (high/low resolution), the type of attack (*whitebox*/*blackbox* attack), and the basis of the method (optimization/learning-based).

Table 1: Comparison with related works.

| Reference | Resolution | White/Black-box | Basis | Available code |
|-----------|-----------|-----------------|-------|----------------|
| Zhmoginov & Sandler (2016) | low | whitebox | 1) optimization 2) learning | ✗ |
| Cole et al. (2017) | low | both* | learning | ✗ |
| Mai et al. (2018) | low | blackbox | learning | ✓ |
| Duong et al. (2020) | low | both** | learning | ✗ |
| Truong et al. (2022) | low | both** | learning | ✗ |
| Dong et al. (2021) | high | blackbox | learning | ✓ |
| Vendrow & Vendrow (2021) | high | blackbox | optimization | ✓ |
| Dong et al. (2022) | high | blackbox | optimization | ✗ |
| Ours | high | both*** | learning | ✓ |

*The method is based on the *whitebox* attack, and is extended to *blackbox* by removing a loss term that required the FR model.
**The method is based on the *whitebox* attack, and the *blackbox* attack is performed by knowledge distillation of the FR model.
***The method is based on the *whitebox* attack, and is extended to *blackbox* using a different FR model.

Zhmoginov & Sandler (2016) proposed an optimization-based method and a learning-based method to generate low-resolution face images in the *whitebox* attack against FR systems. In their optimization-based attack, they used a gradient-descent-based approach to find an image that minimizes the distance of the face template as well as some regularization terms to generate a smooth image, including the total variation and Laplacian pyramid gradient normalization (Burt & Adelson, 1987) of the reconstructed face image. In their learning-based attack, they trained a convolutional neural network (CNN) with the same loss terms to generate face images from given facial templates.

Cole et al. (2017) proposed a learning-based attack to generate low-resolution images using a multi-layer perceptron (MLP) to estimate landmark coordinates and a CNN to generate face textures, and then reconstructed face images using a differentiable warping based on estimated landmarks and face texture. They trained their networks in an end-to-end fashion, and minimized the errors for landmark estimation and texture generation as well as the distance of face template as their loss function. To extend their method from the *whitebox* attack to the *blackbox* attack, they proposed not to minimize the distance of face templates in their loss function.

Mai et al. (2018) proposed a learning-based attack to generate low-resolution images in the *blackbox* attack against FR systems. They proposed new convolutional blocks, called neighborly deconvolution blocks A/B (shortly, NbBlock-A and NbBlock-B), and used these blocks to reconstruct face images. They trained their proposed networks using two loss functions, including pixel loss (i.e., $\ell_2$ norm of reconstruction pixel error) and perceptual loss (i.e., $\ell_2$ norm of distance for intermediate features of VGG-19 (Simonyan & Zisserman, 2014) given original and reconstructed face images).

Duong et al. (2020) and Truong et al. (2022) used a same bijection learning framework and trained a GAN with a generator with structure of PO-GAN (Karras et al., 2017) and TransGAN (Jiang et al., 2021), respectively. While their method is based on the *whitebox* attack, they proposed to use knowledge distillation to extend to the *blackbox* attack. To this end, they trained a student network that mimics the target FR model. However, they did not provide any details (nor source code) about student network training, such as the structure of the student network, etc.

Dong et al. (2021) used a pre-trained StyleGAN to generate high-resolution face images in the *blackbox* attack against FR systems. They generated synthetic face images using pre-trained StyleGAN and extracted their embedding. Then, they trained a fully connected network using mean squared error to map extracted embeddings to the corresponding noise in the input of StyleGAN. Instead of a learning-based approach, Vendrow & Vendrow (2021) used a grid search optimization using the simulated annealing (Van Laarhoven & Aarts, 1987) approach to find the noise in the input of StyleGAN, which generates an image that has the same embedding. As their iterative method has a large computation cost, they evaluated their method on 20 images only. Along the same lines, Dong et al. (2022) also tried to solve a similar optimization to (Vendrow & Vendrow, 2021) with a different approach. They used the genetic algorithm to find the noise in the input of StyleGAN that can generate an image with the same embedding.

Compared to most works in the literature that generate low-resolution face images, our proposed method generates high-resolution realistic face images. While low-resolution reconstructed images

can be used for evaluating the vulnerability of FR systems under some assumptions, high-resolution images can lead to different types of presentation attacks against FR systems. We also propose our method for both *whitebox* and *blackbox* scenarios and evaluate the transferability of our attack. Similar to (Cole et al., 2017; Duong et al., 2020; Truong et al., 2022), our method is based on the *whitebox* knowledge of FR model, however our approach for extending our method to the *blackbox* attack using another FR model is novel. Last but not least, we define five different attacks against FR systems and evaluate the vulnerability of SOTA FR models to our attacks.

## 4 EXPERIMENTS

In this section, we present our experiments and discuss our results. First, in Section 4.1 we describe our experimental setup. Then, we present our experimental results in Section 4.2 and discuss our findings.

### 4.1 EXPERIMENTAL SETUP

To evaluate the performance of our method, we consider two SOTA FR models, including ArcFace (Deng et al., 2019), ElasticFace (Boutros et al., 2022), as the models from which templates are leaked (i.e., $F_{\text{database}}$). For transferability evaluation, we also use three different FR models with SOTA backbones from FaceX-Zoo (Wang et al., 2021), including HRNet (Wang et al., 2020), Attention-Net (Wang et al., 2017), and Swin (Liu et al., 2021), for the target FR system (i.e., $F_{\text{target}}$). The recognition performance of these models are reported in Table 2. All

Table 2: Recognition performance of face recognition models used in our experiments in terms of true match rate (TMR) at the thresholds correspond to false match rates (FMRs) of $10^{-2}$ and $10^{-3}$ evaluated on the MOBIO and LFW datasets. The values are in percentage.

| model | MOBIO | | LFW | |
|---|---|---|---|---|
| | FMR=$10^{-2}$ | FMR=$10^{-3}$ | FMR=$10^{-2}$ | FMR=$10^{-3}$ |
| **ArcFace** | 100.00 | 99.98 | 97.60 | 96.40 |
| **ElasticFace** | 100.00 | 100.00 | 96.87 | 94.70 |
| **HRNet** | 98.98 | 98.23 | 89.30 | 78.43 |
| **AttentionNet** | 99.71 | 97.73 | 84.27 | 72.77 |
| **Swin** | 99.75 | 98.98 | 91.70 | 87.83 |

these models are trained on MS-Celeb1M dataset (Guo et al., 2016). We assume that the adversary does not have access to the FR training dataset, and therefore we use another dataset for training our face reconstruction models. To this end, we use the Flickr-Faces-HQ (FFHQ) dataset (Karras et al., 2019), which consists of 70,000 high-resolution (i.e., $1024 \times 1024$) face images (without identity labels) crawled from the internet. We use 90% random portion of this dataset for training, and the remaining 10% for validation.

To evaluate different attacks against FR systems, we consider two other face image datasets with identity labels, including the MOBIO (McCool et al., 2013) and Labeled Faces in the Wild (LFW) (Huang et al., 2007) datasets. The MOBIO dataset consists of bi-modal (face and voice) data captured using mobile devices from 150 people in 12 sessions (6-11 samples in each session). The LFW dataset includes 13,233 face images of 5,749 people collected from the internet, where 1,680 people have two or more images.

For each of the attacks described in Section 2, we build one or two separate FR systems with one or two SOTA FR models based on the attack type. If the target system is the *same* as the system from which the template is leaked, we have only one FR system. Otherwise, if the target system is *different* the system from which the template is leaked, we have two FR systems with two different feature extractors. In each case, we use one of our evaluation datasets (i.e., MOBIO and LFW) to build both FR systems (so that the subject with the leaked template be enrolled in the target system too). In each evaluation, we assume that the target FR system is configured at the threshold corresponding to a false match rate (FMR) of $10^{-3}$, and we evaluate the adversary's success attack rate (SAR) in entering that system.

We should note that the templates extracted by the aforementioned FR models have 512 dimensions. The input noise $z \in \mathcal{Z}$ to the mapping network of StyleGAN's pre-trained network is from the standard normal distribution and has 512 dimensions. The input noise $n \in \mathcal{N}$ to our mapping network $M_{\text{rec}}(.)$ is with dimension of 8 and also from the standard normal distribution. We also use Adam (Kingma & Ba, 2015) optimizer to train our mapping network.

Table 3: Evaluation of attacks with *whitebox* knowledge of the system from which the template is leaked (i.e., $F_{\text{loss}} = F_{\text{database}}$) against SOTA FR models in terms of adversary's success attack rate (SAR) using our proposed method on the MOBIO and LFW datasets. The values are in percentage and correspond to the threshold where the target system has FMR $= 10^{-3}$. Cells are color coded according the type of attack as defined in Section 2 for attack 1 ( light gray ) and attack 2 ( dark gray ).

| $F_{\text{database}}$ | MOBIO | | | | | LFW | | | | |
|---|---|---|---|---|---|---|---|---|---|---|
| | ArcFace | ElasticFace | HRNet | AttentionNet | Swin | ArcFace | ElasticFace | HRNet | AttentionNet | Swin |
| **ArcFace** | 92.38 | 81.90 | 71.43 | 70.48 | 74.29 | 86.82 | 74.20 | 36.57 | 36.40 | 58.86 |
| **ElasticFace** | 78.10 | 87.62 | 64.29 | 64.76 | 69.05 | 78.25 | 82.52 | 41.80 | 40.25 | 61.09 |

Table 4: Evaluation of attacks (with *blackbox* knowledge of the system from which the template is leaked i.e., $F_{\text{database}}$) against SOTA FR models in terms of adversary's success attack rate (SAR) using different methods on the MOBIO and LFW datasets. The values are in percentage and correspond to the threshold where the target system has FMR $= 10^{-3}$. **M1**: NbNetB-M (Mai et al., 2018), **M2**: NbNetB-P (Mai et al., 2018), **M3**: (Dong et al., 2021), and **M4**: (Vendrow & Vendrow, 2021). Cells are color coded according the type of attack as defined in Section 2 for attack 3 ( lightest gray ), attack 4 ( middle dark gray ), and attack 5 ( darkest gray ).

| $F_{\text{database}}$ | $F_{\text{loss}}$ | $F_{\text{target}}$ | MOBIO | | | | | LFW | | | | |
|---|---|---|---|---|---|---|---|---|---|---|---|---|
| | | | M1 | M2 | M3 | M4 | Ours | M1 | M2 | M3 | M4 | Ours |
| **ArcFace** | **ElasticFace** | **ArcFace** | 1.90 | 15.24 | 2.38 | 28.10 | **81.90** | 10.68 | 40.25 | 12.91 | 58.88 | **77.16** |
| | | **ElasticFace** | 1.43 | 11.43 | 4.29 | 15.24 | **73.81** | 8.36 | 34.39 | 6.35 | 29.10 | **68.06** |
| | | **HRNet** | 0.95 | 6.19 | 2.86 | 10.00 | **57.14** | 1.30 | 7.78 | 1.75 | 9.20 | **28.45** |
| | | **AttentionNet** | 0 | 6.67 | 3.33 | 4.29 | **54.29** | 1.33 | 7.17 | 2.29 | 9.17 | **28.87** |
| | | **Swin** | 1.43 | 13.33 | 3.81 | 10.95 | **67.14** | 4.27 | 23.85 | 5.97 | 21.75 | **48.28** |
| **ElasticFace** | **ArcFace** | **ArcFace** | 2.38 | 18.57 | 2.86 | 16.19 | **87.14** | 15.33 | 48.67 | 11.81 | 37.45 | **83.20** |
| | | **ElasticFace** | 3.81 | 43.81 | 4.76 | 43.33 | **89.05** | 21.44 | 58.16 | 11.59 | 52.88 | **83.43** |
| | | **HRNet** | 0.48 | 20.00 | 1.43 | 10.48 | **73.81** | 3.46 | 18.36 | 2.74 | 11.82 | **49.02** |
| | | **AttentionNet** | 1.90 | 18.10 | 3.33 | 9.05 | **71.90** | 2.89 | 16.31 | 2.91 | 10.95 | **46.63** |
| | | **Swin** | 0.95 | 26.19 | 2.86 | 15.24 | **75.24** | 9.22 | 38.79 | 8.26 | 24.62 | **66.89** |

## 4.2 ANALYZE

In this section, we consider SOTA FR models and evaluate the performance of our face reconstruction method in five different attacks described in Section 2. We also explore the effect of our WGAN traning as well as effect of loss terms as our ablation study.

**Whitebox knowledge of $F_{\text{database}}$** For attacks 1-2, the adversary is assumed to have *whitebox* knowledge of the system from which the template is leaked (i.e., $F_{\text{database}}$) and use the same feature extraction model for training (i.e., $F_{\text{loss}}$), thus in such cases $F_{\text{loss}} = F_{\text{database}}$. We considered ArcFace and ElasticFace models and reconstructed face images from the templates extracted by these models in attacks against different FR systems. Table 3 reports the vulnerability of different target systems to our attacks[1] 1-2 in terms of adversary's SAR at the system's FMR of $10^{-3}$. Similar results for the system's FMR of $10^{-2}$ are reported in Table 6 of Appendix. According to these tables, our method achieves considerable SAR against ArcFace and ElasticFace target systems in attack 1. In attack 2, we observe that there is a degradation in SAR with respect to attack 1. However, the reconstructed face images can still be used to enter another target system. Meanwhile, the FR model with a higher recognition accuracy is generally more vulnerable to attack 2. For instance, when ArcFace is considered as $F_{\text{database}}$, we observe that ElasticFace and Swin have the highest SAR as target systems, while there is the same order for their recognition performance in Table 2.

**Blackbox knowledge of $F_{\text{database}}$** For attacks 3-5, the adversary is assumed to have *blackbox* knowledge of the system from which the template is leaked (i.e., $F_{\text{database}}$) and use another feature extraction model for training (i.e., $F_{\text{loss}}$), therefore in such cases $F_{\text{loss}} \neq F_{\text{database}}$. Table 4 compares

---

[1]We should highlight that since there is no *whitebox* method in the literature with available source code (as mentioned in Table 1), we could not compare our proposed method with other *whitebox* methods.

the performance of our method with *blackbox* methods[2] in the literature (Mai et al., 2018; Dong et al., 2021; Vendrow & Vendrow, 2021) for attacks 3-5 in terms of adversary's SAR at system's FMR of $10^{-3}$. Similar results for the system's FMR of $10^{-2}$ are available in Table 7 of Appendix.

As these tables show, our proposed method achieves the highest SAR compared to (Mai et al., 2018; Dong et al., 2021; Vendrow & Vendrow, 2021) against FR systems on the MO-BIO and LFW datasets. In particular, in attack 5 which is the hardest attack, where $F_{\text{database}}$, $F_{\text{loss}}$, and $F_{\text{target}}$ are different, the results show that the target FR system is still vulnerable to our attack. The results of our method for attack 5 also show transferability of our attack to different FR systems. Similar to attack 2, we can also observe that in attack 5, the FR model with a higher recognition accuracy is generally more vulnerable to our attack. Fig. 4 also shows sample face images from the LFW dataset and the reconstructed images using our proposed method from ArcFace templates in different attacks. We should highlight that as show in Fig. 4, the reconstructed face images in attack 1 and attack 2 are the same, but they are used to enter different target FR system. The same holds for the reconstructed face images in attacks 3-5.

**Ablation Study** To evaluate the effect of WGAN in training our mapping network and the effect of each term in our loss function (i.e., Eq. 1), we consider the ArcFace model in the *whitebox* scenario and train different face reconstruction networks with different loss functions. Then, we attack a system with the ArcFace model as a feature extractor (i.e., attack 1) and compare the SARs as reported in Table 5. According to these results, the proposed adversarial training has a significant effect on our face reconstruction method. In other words, the WGAN framework helps our mapping network to learn the distribution of StyleGAN's intermediate latent space to generate face-like images. When we use the WGAN training and based on the results in Table 5, the ID loss has a high impact on the performance of the template inversion model. While the pixel loss by itself does not achieve a good performance, it improves the performance of ID loss in our reconstruction loss function in Eq. 1. This table confirms that the proposed WGAN training and our reconstruction loss function lead to a more successful attack.

**Limitations** Despite the significant performance of our method in terms of success attack rate in all types of attacks reported in Table 3 and Table 4, the reconstructed face images fail to enter the system in some cases. Fig. 5 illustrates sam-

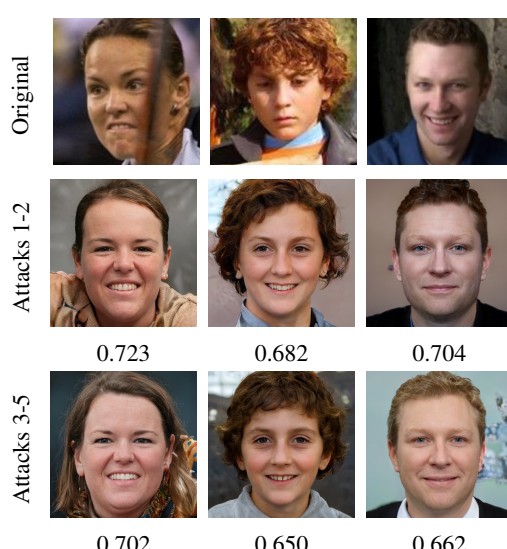

Figure 4: Sample face images from the LFW dataset (first raw) and their corresponding reconstructed images using our template inversion method from ArcFace templates in different attacks, attacks 1-2 (second raw) and attacks 3-5 (second raw, using ElasticFace for $F_{\text{loss}}$). The values below each image show the cosine similarity between the corresponding ArcFace templates of original and reconstructed face images.

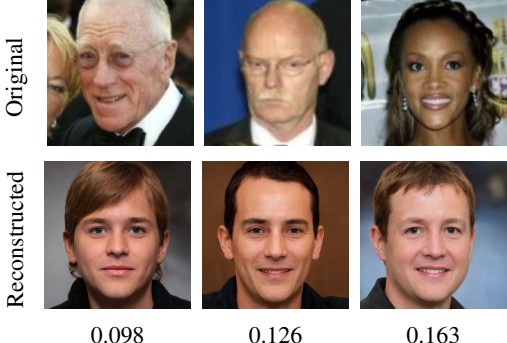

Figure 5: Sample failure cases images from the LFW dataset and their corresponding reconstructed images using our template inversion method from ArcFace templates in the attack 3 (using ElasticFace for $F_{\text{loss}}$). The values below each image show the cosine similarity between the corresponding templates of original and reconstructed face images.

---

[2]The other *blackbox* methods in the literature do not have available source code and we could not reproduce their results.

Table 5: Evaluating the effect of each loss term in our loss function in attack 1 against ArcFace in terms of SAR in the system with FMRs of $10^{-2}$ and $10^{-3}$ evaluated on the MOBIO and LFW datasets. The values are in percentage.

| WGAN training (Eqs. 4 and 5) | Reconstruction Loss Function | MOBIO | | LFW | |
|---|---|---|---|---|---|
| | | FMR=$10^{-2}$ | FMR=$10^{-3}$ | FMR=$10^{-2}$ | FMR=$10^{-3}$ |
| ✓ | $\mathcal{L}_{\text{rec}} = \mathcal{L}_{\text{pixel}} + \mathcal{L}_{\text{ID}}$ | **100.00** | **92.38** | **93.64** | **86.82** |
| | $\mathcal{L}_{\text{rec}} = \mathcal{L}_{\text{ID}}$ | 98.10 | 82.38 | 90.56 | 80.74 |
| | $\mathcal{L}_{\text{rec}} = \mathcal{L}_{\text{pixel}}$ | 0 | 0 | 0.65 | 0.07 |
| ✗ | $\mathcal{L}_{\text{rec}} = \mathcal{L}_{\text{pixel}} + \mathcal{L}_{\text{ID}}$ | 0 | 0 | 0.32 | 0.02 |
| | $\mathcal{L}_{\text{rec}} = \mathcal{L}_{\text{ID}}$ | 0 | 0 | 0.14 | 0.02 |
| | $\mathcal{L}_{\text{rec}} = \mathcal{L}_{\text{pixel}}$ | 0 | 0 | 0.44 | 0.09 |

ple failure cases in the attack 3 against ArcFace (using ElasticFace for $F_{\text{loss}}$) on the LFW dataset. From the failure cases, we can conclude that there is a bias in the face reconstruction for specific demographies, like elderly or dark skin people. Indeed, such kind of bias in the reconstructed face images is caused by inherent biases in datasets used to train FR model, the StyleGAN model, and our mapping network in our face reconstruction model[3].

## 5 CONCLUSION

In this paper, we proposed a new method to reconstruct high-resolution realistic face images from facial templates in a FR system. We used a pre-trained StyleGAN3 network and learned a mapping from facial templates to intermediate latent space of StyleGAN within a GAN-based framework. We proposed our method for *whitebox* and *blackbox* scenarios. In the *whitebox* scenario, the adversary can use the feature extraction model for training the face reconstruction network; however, in the *blackbox* scenario, we assume that the adversary has access to another feature extraction model. In addition, we consider the threat model where the adversary might impersonate in the same or another (i.e., transferable attack) FR system. Based on the adversary's knowledge of the feature extraction model and the target FR system, we defined five different attacks and evaluated the vulnerability of SOTA FR systems to our proposed method. Our experiments showed that the reconstructed face images by our proposed method not only can achieve a high SAR in *whitebox* and *blackbox* scenarios, but also are transferable and can be used to enter target FR systems with a different FR model.

## ETHICS STATEMENT

**Motivations** The proposed face reconstruction method is presented with the motivation of showing vulnerability of face recognition systems to template inversion attacks. We hope this work encourages researcher of the community to investigate the next generation of safe and robust face recognition systems and to develop new algorithms to protect existing systems.

**Considerations** While the proposed method might pose a social threat against unprotected systems, we do not condone using our work with the intent of attacking a *real* face recognition system or other malicious purposes. The authors also acknowledge a potential lack of diversity in the reconstructed face images, stemming from inherent biases of datasets used in our experiments.

**Mitigation of such attacks** This paper demonstrates an important privacy and security threat to the state-of-the-art unprotected face recognition systems. Along the same lines, data protection frameworks, such as the European Union General Data Protection Regulation (EU-GDPR) (European Council, 2016), put legal obligations to protect biometric data as sensitive information. To this end and to prevent such attacks to face recognition systems, several biometric template protection algorithms are proposed in the literature (Nandakumar & Jain, 2015; Sandhya & Prasad, 2017; Kaur et al., 2022; Kumar et al., 2020).

---

[3]The biases for different demographies in verification task for ArcFace model are studied in (de Freitas Pereira & Marcel, 2021). Similarly, biases in StyleGAN generated images and also the FFHQ dataset (i.e., our training dataset) are investigated in (Karakas et al., 2022; Tan et al., 2020; Balakrishnan et al., 2020).

## REPRODUCIBILITY STATEMENT

In our experiments, we use PyTorch package and trained our models on a system equipped with an NVIDIA GeForce RTX$^{\text{TM}}$ 3090. We use the pre-trained model of StyleGAN3[4] to generate $1024 \times 1024$ high-resolution images. The source code of our experiments is publicly available to help reproduce our results[5].

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

# A  APPENDIX

Table 6: Evaluation of attacks with *whitebox* knowledge of the system from which the template is leaked (i.e., $F_{\text{loss}} = F_{\text{database}}$) against SOTA FR models in terms of adversary's success attack rate (SAR) using our proposed method on the MOBIO and LFW datasets. The values are in percentage and correspond to the threshold where the target system has **FMR$= 10^{-2}$**. Cells are color coded according the type of attack as defined in Section 2 for attack 1 ( light gray ) and attack 2 ( dark gray ).

| $F_{\text{database}}$ | MOBIO | | | | | LFW | | | | |
|---|---|---|---|---|---|---|---|---|---|---|
| | ArcFace | ElasticFace | HRNet | AttentionNet | Swin | ArcFace | ElasticFace | HRNet | AttentionNet | Swin |
| **ArcFace** | 100.00 | 93.81 | 80.00 | 81.90 | 85.24 | 93.64 | 90.89 | 68.08 | 62.75 | 76.24 |
| **ElasticFace** | 90.95 | 93.33 | 78.57 | 83.81 | 84.29 | 87.88 | 92.80 | 71.82 | 64.24 | 75.70 |

Table 7: Evaluation of attacks (with *blackbox* knowledge of the system from which the template is leaked i.e., $F_{\text{database}}$) against SOTA FR models in terms of adversary's success attack rate (SAR) using different methods on the MOBIO and LFW datasets. The values are in percentage and correspond to the threshold where the target system has **FMR$= 10^{-2}$**. **M1**: NbNetB-M (Mai et al., 2018), **M2**: NbNetB-P (Mai et al., 2018), **M3**: (Dong et al., 2021), and **M4**: (Vendrow & Vendrow, 2021). Cells are color coded according the type of attack as defined in Section 2 for attack 3 ( lightest gray ), attack 4 ( middle dark gray ), and attack 5 ( darkest gray ).

| $F_{\text{database}}$ | $F_{\text{loss}}$ | $F_{\text{target}}$ | MOBIO | | | | | LFW | | | | |
|---|---|---|---|---|---|---|---|---|---|---|---|---|
| | | | M1 | M2 | M3 | M4 | Ours | M1 | M2 | M3 | M4 | Ours |
| **ArcFace** | **ElasticFace** | **ArcFace** | 26.67 | 49.05 | 20.48 | 67.14 | **89.52** | 26.66 | 61.66 | 28.31 | 76.98 | **87.85** |
| | | **ElasticFace** | 11.90 | 49.52 | 16.19 | 34.29 | **86.67** | 32.42 | 66.61 | 23.05 | 57.84 | **87.43** |
| | | **HRNet** | 10.48 | 24.76 | 10.00 | 26.19 | **79.05** | 18.69 | 43.21 | 17.37 | 33.55 | **60.93** |
| | | **AttentionNet** | 11.43 | 38.10 | 18.10 | 24.29 | **80.48** | 10.84 | 31.88 | 13.31 | 26.73 | **53.86** |
| | | **Swin** | 10.48 | 45.24 | 10.95 | 29.52 | **82.86** | 14.79 | 45.80 | 16.98 | 38.03 | **67.80** |
| **ElasticFace** | **ArcFace** | **ArcFace** | 17.14 | 49.05 | 20.95 | 47.14 | **95.24** | 33.08 | 67.89 | 26.35 | 57.48 | **91.23** |
| | | **ElasticFace** | 30.00 | 70.95 | 25.7 | 75.24 | **94.76** | 52.99 | 81.74 | 33.53 | 79.62 | **93.34** |
| | | **HRNet** | 8.10 | 47.14 | 15.24 | 31.43 | **83.81** | 29.27 | 60.34 | 23.22 | 39.06 | **76.68** |
| | | **AttentionNet** | 12.86 | 47.14 | 23.43 | 40.95 | **87.14** | 18.53 | 46.36 | 17.78 | 31.53 | **69.45** |
| | | **Swin** | 10.00 | 54.76 | 13.81 | 37.14 | **89.05** | 24.50 | 60.19 | 21.40 | 41.13 | **80.15** |

