# OpenReview forum: "Face reconstruction from facial templates by learning latent space of a generator network"
_ICLR.cc/2023/Conference — Submitted to ICLR 2023_

### Official Review · Reviewer_bw96 · 2022-10-23

**Confidence:** 3
**Correctness:** 3
**Technical Novelty And Significance:** 2
**Empirical Novelty And Significance:** 2
**Recommendation:** 6

**Clarity, Quality, Novelty And Reproducibility:**

The paper is clearly writen and easy to read and understand.

The technical novelty is limited in my opinion because the proposed method mainly relies on StyleGAN3 for face reconstruction.

Details for reproducibility are given.

**Details Of Ethics Concerns:**

The paper focuses on the problem of adversarial attacks in face recognition, which has clear implications for privacy and safety. Authors included an ethics statement, but they should also try to illustrate how to mitigate the potential issues deriving from their proposed attack method.
Make public the code of the paper could result in potential risk.


**Strength And Weaknesses:**

The main contributions of the paper are:
-	A method to generate high-resolution realistic face images from facial templates. Within a GAN-based framework, the mapping from facial templates to the latent space of a pre-trained face generation network is learned.
-	The method is proposed for whitebox and blackbox attack scenarios. The method is based on the whitebox knowledge of the face recognition model, and it then extended to an attack blackbox scenario, using another face recognition model that the adversary has access to.

There are some aspects in the paper that need clarification:
-	In my opinion the novelty of the paper is limited. The proposed method mainly relies on StyleGAN3 for face reconstruction.
-	Due to the social security and ethical implications of this works, authors could have described possible solutions to mitigate the effect of the proposed attacks.
-	Authors did not discuss the results in a good way. It is not clear which are the limitations of the method, and which are the case some reconstructions are not well recognized. No examples are given of such cases.


**Summary Of The Paper:**

This paper focuses on template inversion attack against face recognition systems and proposes a method to reconstruct face images from facial templates. In a GAN-based framework, a mapping from facial templates to the intermediate latent space of a pre-trained face generation network is learned, from which high-resolution reconstructed face images can be generated. The proposed method can be
applied in whitebox and blackbox attacks against face recognition systems.


**Summary Of The Review:**

The paper is clearly written and easy to read and understand.

In my evalaution I weighted more the weaknesses and the fact the actual novelty contribution is limited.

---

> ### Author Response · Authors · 2022-11-17
> **Authors Response to Comments of Reviewer bw96 [Part 3/3]**
>
> > [Flag For Ethics Review]: Yes, Privacy, security and safety
> >
> > [Details Of Ethics Concerns] The paper focuses on the problem of adversarial attacks in face recognition, which has clear implications for privacy and safety. Authors included an ethics statement, but they should also try to illustrate how to mitigate the potential issues deriving from their proposed attack method. Make public the code of the paper could result in potential risk.
>
> We tried to address each of these concerns raised by the reviewer, separately:
>
> >The paper focuses on the problem of adversarial attacks in face recognition, which has clear implications for privacy and safety.
>
> The main motivation of the paper is to evaluate the vulnerability of face recognition systems to template inversion attacks. In fact, as also mentioned by the reviewer, such a threat jeopardizes the privacy and security of users in face recognition systems. However, for developing robust and safe face recognition systems, any potential attack against these systems should be investigated. In other words, to increase the robustness of any system, any potential threats against that system should be studied.
> In addition to the necessity of such a study, we should note that the project on which the work has been conducted has passed an Internal Ethical Review Board (IRB).
>
> > Authors included an ethics statement, but they should also try to illustrate how to mitigate the potential issues deriving from their proposed attack method.
>
> We thank the reviewer for the suggestion. As mentioned in an earlier answer, we revised the "Ethics Statement" section and added a new paragraph explaining how to mitigate such kinds of attacks. In summary, to prevent such attacks against face recognition systems, several biometric template protection algorithms are proposed in the literature. Data protection regulations (such as EU-GDPR) also pose legal obligations to protect biometric templates. We also added citations to some references for biometric template protection algorithms to help readers with the ways to mitigate such attacks.
>
> > Make public the code of the paper could result in potential risk.
>
> While we understand the reviewer's concerns about the potential misuse of our work, we should highlight that this work is presented with the motivation of showing the vulnerability of face recognition systems to template inversion attacks. In fact, recent data protection frameworks (such as EU-GDPR) put legal obligations to protect the biometric templates.
> The results in our paper demonstrate an important potential threat in unprotected face recognition systems. However, we believe that for the next generation of safe face recognition systems, any kind of potential attacks should be completely studied by the researchers, and then based upon such vulnerability studies, new protection and defense algorithms will be proposed by the research community in the future. To this end, reproducibility is necessary, and therefore, to help future researchers and facilitate the reproducibility of our work, we believe we need to publish the source codes of our experiments.

---

> ### Author Response · Authors · 2022-11-17
> **Authors Response to Comments of Reviewer bw96 [Part 2/3]**
>
> > - Due to the social security and ethical implications of this works, authors could have described possible solutions to mitigate the effect of the proposed attacks.
>
> We thank the reviewer for the suggestion. We revised the "Ethics Statement" section and added a new paragraph explaining how to mitigate such kinds of attacks. In summary, to prevent such attacks against face recognition systems, several biometric template protection algorithms are proposed in the literature. Data protection regulations (such as EU-GDPR) also pose legal obligations to protect biometric templates. We added citations to some references for biometric template protection algorithms to provide readers with references to ways to mitigate such attacks.
>
>
> > - Authors did not discuss the results in a good way. It is not clear which are the limitations of the method, and which are the case some reconstructions are not well recognized. No examples are given of such cases.
>
> We thank the reviewer for the comment. We added a "Limitations" section for the cases where the reconstructed face images fail to enter the system. We discussed the biases in datasets used for training the face recognition model, StyleGAN model, and our mapping network. We also added sample images for such failure cases.
>
> In addition to the discussed limitations, we added further experiments and discussed the results further. In particular,
> - We extended our ablation study and included the cases where WGAN is not used during training. The new ablation study sheds light on the effect of our WGAN learning. In fact, without using adversarial learning, the mapping network easily diverges and generates intermediate latent vectors which are out of distribution. Therefore, the outputs of StyleGAN's generator will not look like a face.
> - We evaluated two other baselines for blackbox attacks (M2 and M4 in Table 4). The new results confirm that our method still outperforms other methods in the literature.
> - We added new sample reconstructed face images for different attacks against ArcFace and discussed the results.
>
> In addition to the new experiments and analyses, we should also highlight that since we will publish the source code of our experiments, researchers can reproduce our results and further explore our proposed method.
>
> > The technical novelty is limited in my opinion because the proposed method mainly relies on StyleGAN3 for face reconstruction.
> > In my evalaution I weighted more the weaknesses and the fact the actual novelty contribution is limited.
>
> As explained in the answer to the previous comment from the reviewer, we believe the contributions of the paper are in two aspects, 1) proposing a new method to reconstruct face images and 2) defining 5 different attacks and evaluating the vulnerability of SOTA models. We should note that there are several other works in the literature for face reconstruction using StyleGAN too (Dong et al., 2021; 2022; Vendrow & Vendrow, 2021). However, the results in Table 4 show that our method far outperforms previous StyleGAN-based face reconstruction methods (M3 and M4 in Table 4 of the revised version of the manuscript). Therefore, although our method inherits the face generation capability of StyleGAN, our proposed method to learn the mapping network from facial templates to the intermediate latent space of StyleGAN is novel and leads to high success attack rates. Furthermore, our experiments show that SOTA face recognition models are vulnerable to our proposed attack under different conditions. We believe that in addition to all the mentioned contributions, this paper motivates researchers to develop more secure face recognition models.

---

> ### Author Response · Authors · 2022-11-17
> **Authors Response to Comments of Reviewer bw96 [Part 1/3]**
>
> We thank the reviewer for their valuable comments. We are happy that the reviewer found our paper well-written. We appreciate the reviewer's comments on the three strengths of our work.
> Below, we tried to address the concerns raised by the reviewer. We hope that it will encourage discussion if any points remain unclear.
>
> > There are some aspects in the paper that need clarification:
> > - In my opinion the novelty of the paper is limited. The proposed method mainly relies on StyleGAN3 for face reconstruction.
>
> We believe the contributions of the paper are in two aspects:
> 1. **Proposing a new method to reconstruct face images:** To reconstruct face images from facial templates, we use a pre-trained StyleGAN and learn a mapping from facial templates to the intermediate latent space of StyleGAN. Then, we use the generator of StyleGAN to generate face images. We learn the intermediate latent space using a GAN-based framework. Compared to the works in the literature, our proposed method is novel, because:
>     - [**First work to learn the intermediate layer**] We learn the mapping from facial templates to the intermediate layer of StyleGAN. While some other works in the literature also use StyleGAN for face reconstruction (Dong et al., 2021; 2022; Vendrow & Vendrow, 2021), none of them learn the intermediate layer. In the previous works, the input noise to the StyleGAN is learned/optimized to generate a reconstructed face image. However, learning the intermediate layer is not straightforward (and as mentioned earlier it has not been proposed in the literature).
>     - [**Use a GAN-based framework for training our mapping network**] To learn the intermediate layer of StyleGAN, we use a GAN-based framework. In the revised version of the manuscript, we added an ablation study to show the effect of our GAN-based learning. In a nutshell, as results show without using the WGAN framework the mapping network will easily diverge and generate intermediate latent vectors out of distribution. Therefore, our GAN-based training of the mapping network helps our model to generate latent codes in the StyleGAN's distribution, and therefore can be used to generate realistic face images.
>     - [**High Success Attack Rates**] The success attack rates from our method compared to other methods show that our method far outperforms other works in the literature, which confirms the effect of our ideas for learning the intermediate layer of StyleGAN from facial templates.
>
> 2. **Defining 5 different attacks and evaluating the vulnerability of SOTA models:** In this paper, we consider both whitebox and blackbox scenarios and we define 5 different attacks against face recognition systems based on the adversary's knowledge. To our knowledge, these five different attacks against face recognition systems are novel and have not been studied in the literature. We believe such a comprehensive definition and evaluation of different attacks against face recognition systems will open new directions for robustness improvement and vulnerability evaluation of face recognition systems.
>
> In addition to the above points, we should note that although our method is based on StyleGAN, it is realistic to assume that the adversary has access to such an open-source face generator network, that is also independent of the face recognition model. Furthermore, there is the same assumption in several SOTA methods proposed in the literature (Dong et al., 2021; 2022; Vendrow & Vendrow, 2021).  It is also worth mentioning that final reconstructed face images have high-resolution (i.e., 1024x1024), and therefore can be used for practical attacks against face recognition models.
>
> We should note that there are several other works in the literature for face reconstruction using StyleGAN too (Dong et al., 2021; 2022; Vendrow & Vendrow, 2021). However, the results in Table 4 show that our method far outperforms previous StyleGAN-based face reconstruction methods (M3 and M4 in Table 4 of the revised version of the manuscript). Therefore, although our method inherits the face generation capability of StyleGAN, our proposed method to learn the mapping network from facial templates to the intermediate latent space of StyleGAN is novel and leads to high success attack rates.

---

### Official Review · Reviewer_dNW2 · 2022-10-23

**Confidence:** 5
**Correctness:** 3
**Technical Novelty And Significance:** 3
**Empirical Novelty And Significance:** 3
**Recommendation:** 6

**Clarity, Quality, Novelty And Reproducibility:**

Clarity
Low

Quality
Good

Novelty
Medium

Reproducibility
Medium

**Details Of Ethics Concerns:**

It was better, not to mentioned to the github link in this stage or make it blind!

**Strength And Weaknesses:**

Strength
1.using a pre-trained StyleGAN3 network and learned a mapping from facial templates to intermediate latent space of StyleGAN within a GAN-based framework
2. proposed method for whitebox and blackbox scenarios
3. defined five different attacks and evaluated the vulnerability of SOTA FR systems to our proposed method.

Weaknesses
1. More details about the method
2. More details about the attacks
3. More analysis on white and black box


**Summary Of The Paper:**

In this paper, they focus on the template inversion attack against face recognition systems. Within a generative adversarial network (GAN)-based framework,  learn a mapping from facial templates to the intermediate latent space of a pre-trained face generation network. their proposed method achieves high success attack rates in white box and black box scenarios.

**Summary Of The Review:**

The paper presented a method for better resolution using Style GAN.
Its structure and methods looks good and novel, however, more details in mythology and presentation is needed.

---

> ### Author Response · Authors · 2022-11-17
> **Authors Response to Comments of Reviewer dNW2**
>
> We thank the reviewer for their valuable comments. We are happy that the reviewer found our paper well-written. We appreciate the reviewer's comments on the three strengths of our work.
> Below, we tried to address the concerns raised by the reviewer:
>
> > Weaknesses:
> > - More details about the method
> > - More details about the attacks
> > - More analysis on white and black box
>
> We thank the reviewer for the comment. As suggested by the reviewer, we added more explanations and details to the paper for the proposed method and our defined attacks in Section 2 of the revised manuscript. Furthermore, several implementation details are also explained at the end of the experimental setup (i.e., Section 4.1.).
>
> Regarding more analyses suggested by the reviewer, we added several new analyses to the paper:
> - We evaluated two other baselines for blackbox attacks (M2 and M4 in Table 4). The new results confirm that our method still outperforms other methods in the literature. The other remaining blackbox methods in the literature do not have available source code and we could not reproduce their results. In the case of the whitebox scenario, none of the whitebox methods in the literature (as mentioned in Table 1) has available source code.
> - As a new analysis of our proposed method, we extended our ablation study and evaluated the case where WGAN training is not used. The new experiments clearly show the effect of adversarial learning in our proposed method.
> - We added new sample reconstructed face images for different attacks against ArcFace and discussed the results.
> - We added a "Limitations" section for the cases where the reconstructed face images fail to enter the system. We discussed the biases caused by datasets used for training the face recognition model, StyleGAN model, and our mapping network. We also added sample images for such failure cases.
>
>
> > The paper presented a method for better resolution using Style GAN. Its structure and methods looks good and novel, however, more details in mythology and presentation is needed.
>
> We thank the reviewer for the comment. As descibed in the answer to the previous comment and according to the reviewer's suggestion, we added more explanations and details to the paper.
> In addition to the newly added details, we should also highlight that since we will publish the source code of our experiments, researchers can use our source code for any missing detail and for further experiments.
>
> > [Flag For Ethics Review]: Yes, Privacy, security and safety
>
> The main motivation of the paper is to evaluate the vulnerability of face recognition systems to template inversion attacks. In fact, a template inversion attack jeopardizes the privacy and security of users in face recognition systems. However, for developing robust and safe face recognition systems, any potential attack against these systems should be investigated. In other words, to increase the robustness of any system, any potential threats against that system should be studied.
> In the revised version of the manuscript, we revised the "Ethics Statement" section and discussed our motivation and ethical considerations. We also added a new paragraph explaining how to mitigate such kinds of attacks. In summary, to prevent such attacks against face recognition systems, several biometric template protection algorithms are proposed in the literature. Data protection regulations (such as EU-GDPR) also pose legal obligations to protect biometric templates. We also added citations to some references for biometric template protection algorithms to help readers with the ways to mitigate such attacks.
>
> In addition to the necessity of such a study and the explanations in the ethical statement, we should note that the project on which the work has been conducted has passed an Internal Ethical Review Board (IRB).
>
>
> > It was better, not to mentioned to the github link in this stage or make it blind!
>
> The *only* provided GitHub link in the submitted manuscript is for a previously published paper in the literature (i.e., StyleGAN3), and is mentioned to cite the implementation of that work.

---

### Official Review · Reviewer_3DHW · 2022-10-24

**Confidence:** 4
**Correctness:** 4
**Technical Novelty And Significance:** 3
**Empirical Novelty And Significance:** 4
**Recommendation:** 6

**Clarity, Quality, Novelty And Reproducibility:**

The paper is well written. The results are new, relevant and they can be reproduced. The authors promise to make publicly available the experimentation code.

**Strength And Weaknesses:**

Strength.

The paper addresses a relevant problem and proves an important vulnerability of present FR systems. It is well written and the experimentation seems correct.

Weaknesses.

My main complain is the title of the paper, that, in my opinion, is misleading. The main contribution in the paper is using results in the literature to engineer of a set of attacks of an inrceasing degree of difficulty proving in all of them the vulnerability of present FR systems.

The face generation part is also relevant, but I would attribute the excellent performance of the attacks to StyleGAN's merits. From the experimentation in the paper it is not clear what is the contribution of the adversarial training. In your ablation you only analyze the contribution of (2) and (3), what about (4) and (5)?


**Summary Of The Paper:**

The paper studies the template inversion attack against face recognition (FR) systems. It uses results in the literature to train an encoder that generates the latent representation of StyleGAN and, using StyleGAN decoder, produces a realistic face, that is subsequently used to attack a FR system.


**Summary Of The Review:**

The paper proves how vulnerable are present FR systems to the template inversion attack even if the intruder does not know the model used to produce the template nor the FR algorithm.

---

> ### Author Response · Authors · 2022-11-17
> **Authors Response to Comments of Reviewer 3DHW [Part 2/2]**
>
> > The face generation part is also relevant, but I would attribute the excellent performance of the attacks to StyleGAN's merits.
>
> As mentioned in the response to the previous comment, we should highlight that there are several other works in the literature for face reconstruction using StyleGAN too (Dong et al., 2021; 2022; Vendrow & Vendrow, 2021). However, the results in Table 4 show that our method far outperforms previous StyleGAN-based face reconstruction methods (M3 and M4 in Table 4 of the revised version of the manuscript). Therefore, although our method inherits the face generation capability of StyleGAN, our proposed method to learn the mapping network from facial templates to the intermediate latent space of StyleGAN is novel and leads to high success attack rates.
>
>
> > From the experimentation in the paper it is not clear what is the contribution of the adversarial training. In your ablation you only analyze the contribution of (2) and (3), what about (4) and (5)?
>
> We thank the reviewer for this comment. As suggested by the reviewer, we extended our ablation study and included the cases where WGAN is not used during training. The new ablation study sheds light on the effect of our WGAN learning. In fact, without using adversarial learning, the mapping network easily diverges and generates intermediate latent vectors which are out of distribution. Therefore, the outputs of StyleGAN's synthesis part will not look like a face.
>
> In addition to the new ablation study which clearly shows the effect of adversarial training, we should also highlight that since we will publish the source code of our experiments, researchers can explore further the effect of any other parts of our proposed method.

---

> ### Author Response · Authors · 2022-11-17
> **Authors Response to Comments of Reviewer 3DHW [Part 1/2]**
>
> We thank the reviewer for their valuable comments. We are happy that the reviewer found our paper well-written. We appreciate the reviewer's comments on acknowledging that *all the claims and statements are well-supported and correct*. We also thank the reviewer for confirming that the contributions in terms of "Empirical Novelty And Significance" are *significant, and do not exist in prior works*.
> Below, we tried to address the concerns raised by the reviewer:
>
>
> > My main complain is the title of the paper, that, in my opinion, is misleading. The main contribution in the paper is using results in the literature to engineer of a set of attacks of an inrceasing degree of difficulty proving in all of them the vulnerability of present FR systems.
>
> We believe the contributions of the paper are in two aspects:
> 1. **Proposing a new method to reconstruct face images:** To reconstruct face images from facial templates, we use a pre-trained StyleGAN and learn a mapping from facial templates to the intermediate latent space of StyleGAN. Then, we use the generator of StyleGAN to generate face images. We learn the intermediate latent space using a GAN-based framework. Compared to the works in the literature, our proposed method is novel, because:
>     - [**First work to learn the intermediate layer**] We learn the mapping from facial templates to the intermediate layer of StyleGAN. While some other works in the literature also use StyleGAN for face reconstruction (Dong et al., 2021; 2022; Vendrow & Vendrow, 2021), none of them learn the intermediate layer. In the previous works, the input noise to the StyleGAN is learned/optimized to generate a reconstructed face image. However, learning the intermediate layer is not straightforward (and as mentioned earlier it has not been proposed in the literature).
>     - [**Use a GAN-based framework for training our mapping network**] To learn the intermediate layer of StyleGAN, we use a GAN-based framework. In the revised version of the manuscript, we added an ablation study to show the effect of our GAN-based learning. In a nutshell, as results show without using the WGAN framework the mapping network will easily diverge and generate intermediate latent vectors out of distribution. Therefore, our GAN-based training of the mapping network helps our model to generate latent codes in the StyleGAN's distribution, and therefore can be used to generate realistic face images.
>     - [**High Success Attack Rates**] The success attack rates from our method compared to other methods show that our method far outperforms other works in the literature, which confirms the effect of our ideas for learning the intermediate layer of StyleGAN from facial templates.
>
> 2. **Defining 5 different attacks and evaluating the vulnerability of SOTA models:** In this paper, we consider both whitebox and blackbox scenarios and we define 5 different attacks against face recognition systems based on the adversary's knowledge. To our knowledge, these five different attacks against face recognition systems are novel and have not been studied in the literature. We believe such a comprehensive definition and evaluation of different attacks against face recognition systems will open new directions for robustness improvement and vulnerability evaluation of face recognition systems.
>
>
> We should note that there are several other works in the literature for face reconstruction using StyleGAN too (Dong et al., 2021; 2022; Vendrow & Vendrow, 2021). However, the results in Table 4 show that our method far outperforms previous StyleGAN-based face reconstruction methods (M3 and M4 in Table 4 of the revised version of the manuscript). Therefore, although our method inherits the face generation capability of StyleGAN, we respectfully disagree with the reviewer and rather claim that **our proposed method to learn the mapping network from facial templates to the intermediate latent space of StyleGAN is novel and leads to high success attack rates**.
> Furthermore, the second contribution of defining 5 different attacks and evaluating the vulnerability of SOTA models against these attacks is also a novel evaluation (which was not investigated in the literature to the best of our knowledge) to study the effect of our proposed attack. Considering our defined attacks and as experimental results show, our proposed method outperforms previous methods in the literature in all different types of attacks. Furthermore, the results show the considerable vulnerability of SOTA face recognition models to our attack under different conditions. Therefore, we still believe the main contribution of the paper is our face reconstruction method which is represented in the title of the paper.

---

### Official Review · Reviewer_e7hV · 2022-10-25

**Confidence:** 4
**Correctness:** 4
**Technical Novelty And Significance:** 3
**Empirical Novelty And Significance:** 3
**Recommendation:** 6

**Clarity, Quality, Novelty And Reproducibility:**

- The novelty and scope appear limited and incremental. The main contribution in the paper is in using results in StyleGAN in  FR systems.
- The authors promise to make publicly available the experimentation code.


**Details Of Ethics Concerns:**

The authors should explain how to mitigate the potential issues deriving from the proposed attack method.

**Strength And Weaknesses:**


Strength
- The motivation is very clear.
-  It is well written and the experimentation seems correct.
- Five different attacks and evaluated the vulnerability of SOTA FR systems to the proposed method are defined.


**Summary Of The Paper:**

The authors of this paper propose  new method to reconstruct high-resolution realistic face images from
facial templates in a FR system. They focus on the template inversion attack against face recognition systems.

**Summary Of The Review:**

The paper is clearly written and easy to read and understand. But the novelty and scope appear limited.

---

> ### Author Response · Authors · 2022-11-17
> **Authors Response to Comments of Reviewer e7hV [Part 2/2]**
>
> > The authors should explain how to mitigate the potential issues deriving from the proposed attack method.
>
> We thank the reviewer for the suggestion. We revised the "Ethics Statement" section and added a new paragraph explaining how to mitigate such kinds of attacks. In summary, to prevent such attacks against face recognition systems, several biometric template protection algorithms are proposed in the literature. Data protection regulations (such as EU-GDPR) also pose legal obligations to protect biometric templates. We also added citations to some references for biometric template protection algorithms to provide readers with references to ways to mitigate such attacks.
>
> > Flag For Ethics Review: Yes, Legal compliance (e.g., GDPR, copyright, terms of use)
>
> We have signed the licenses (GDPR compliance) to use from the data controller of any of the datasets used in this paper (i.e., MOBIO, LFW, and FFHQ). We are following the terms of use of these datasets. All in all, considering the signed licenses and terms of use, we believe there is no copyright issue.
>
> In addition, we should note that the project on which the work has been conducted has passed an Internal Ethical Review Board (IRB).

---

> ### Author Response · Authors · 2022-11-17
> **Authors Response to Comments of Reviewer e7hV [Part 1/2]**
>
> We thank the reviewer for their valuable comments. We are happy that the reviewer found our paper well-written with very clear motivation. We appreciate the reviewer's comments on acknowledging that *all of the claims and statements are well-supported and correct*.
> Below, we tried to answer the concerns raised by the reviewer:
>
> > The novelty and scope appear limited and incremental. The main contribution in the paper is in using results in StyleGAN in FR systems.
>
> The contributions of the paper are in two aspects:
> 1. **Proposing a new method to reconstruct face images:** To reconstruct face images from facial templates, we use a pre-trained StyleGAN and learn a mapping from facial templates to the intermediate latent space of StyleGAN. Then, we use the generator of StyleGAN to generate face images. We learn the intermediate latent space using a GAN-based framework. Compared to the works in the literature, our proposed method is novel, because:
>     - [**First work to learn the intermediate layer**] We learn the mapping from facial templates to the intermediate layer of StyleGAN. While some other works in the literature also use StyleGAN for face reconstruction (Dong et al., 2021; 2022; Vendrow & Vendrow, 2021), none of them learn the intermediate layer. In the previous works, the input noise to the StyleGAN is learned/optimized to generate a reconstructed face image. However, learning the intermediate layer is not straightforward (and as mentioned earlier it has not been proposed in the literature).
>     - [**Use a GAN-based framework for training our mapping network**] To learn the intermediate layer of StyleGAN, we use a GAN-based framework. In the revised version of the manuscript, we added an ablation study to show the effect of our GAN-based learning. In a nutshell, as results show without using the WGAN framework the mapping network will easily diverge and generate intermediate latent vectors out of distribution. Therefore, our GAN-based training of the mapping network helps our model to generate latent codes in the StyleGAN's distribution, and therefore can be used to generate realistic face images.
>     - [**High Success Attack Rates**] The success attack rates from our method compared to other methods show that our method far outperforms other works in the literature, which confirms the effect of our ideas for learning the intermediate layer of StyleGAN from facial templates.
>
> 2. **Defining 5 different attacks and evaluating the vulnerability of SOTA models:** In this paper, we consider both whitebox and blackbox scenarios and we define 5 different attacks against face recognition systems based on the adversary's knowledge. To our knowledge, these five different attacks against face recognition systems are novel and have not been studied in the literature. We believe such a comprehensive definition and evaluation of different attacks against face recognition systems will open new directions for robustness improvement and vulnerability evaluation of face recognition systems.
>
> In addition to the above points, we should note that although our method is based on StyleGAN, it is realistic to assume that the adversary has access to such an open-source face generator network, that is also independent of the face recognition model. Furthermore, there is the same assumption in several SOTA methods proposed in the literature
> (Dong et al., 2021; 2022; Vendrow & Vendrow, 2021).  It is also worth mentioning that final reconstructed face images have high-resolution (i.e., 1024x1024), and therefore can be used for practical attacks against face recognition models.
>
> We should note that there are several other works in the literature for face reconstruction using StyleGAN too
> (Dong et al., 2021; 2022; Vendrow & Vendrow, 2021). However, the results in Table 4 show that our method far outperforms previous StyleGAN-based face reconstruction methods (M3 and M4 in Table 4 of the revised version of the manuscript). Therefore, although our method inherits the face generation capability of StyleGAN, our proposed method to learn the mapping network from facial templates to the intermediate latent space of StyleGAN is novel and leads to high success attack rates.

---

### Author Response · Authors · 2022-11-17
**Authors General Response to Reviewers of Paper6309**

We thank all reviewers for their time and valuable comments. We are glad to receive positive feedback from reviewers, particularly:
- **[all reviewers]** All reviewers found the paper well-written and easy to read and understand.
- **[all reviewers]** R3 and R4 acknowledged the reproducibility of our work, and in addition, R1 and R2 noted reproducibility based on publishing source codes (as mentioned in the paper, we will publish the source code of all experiments upon acceptance of the paper).
- **[R1 and R2]** R1 and R2 found *all of the claims and statements well-supported and correct*.
- **[R3 and R4]** R3 and R4 found one of the strengths of our paper is that our method is proposed for both whitebox and blackbox scenarios.
- **[R1 and R3]** R1 and R3 found one of the strengths of our paper is that we defined five different attacks and evaluated the vulnerability of SOTA FR systems to our proposed method.
- **[R1 and R3]** R3 acknowledged learning a mapping from facial templates to intermediate latent space of StyleGAN within a GAN-based framework as a strength of our work. Also, R1 found using StyleGAN as a contribution of our work.
- **[R2]** R2 found the contributions in terms of *"Empirical Novelty And Significance"* to be *"significant, and do not exist in prior works."*
- **[R1]** R1 found the motivation of the paper very clear.

In addition to the above comments, we received valuable feedback from the reviewers which helps us improve the quality of the paper. We implemented several new experiments according to the comments and improved the manuscript with further discussions and explanations. We tried to address every point raised by reviewers in the individual responses.
The summary of changes to the revised version of the paper is as follows:
- **[New ablation study]** **[R2 and R3]** We extended our ablation study and evaluated the cases where WGAN training is not used. The new experiments clearly show the effect of adversarial learning in our proposed method. (as suggested by R2 and as another analysis suggested by R3)
- **[New limitations section]** **[R3 and R4]** We added a "Limitations" section for the cases where the reconstructed face images fail to enter the system. We discussed the biases caused by datasets used for training the face recognition model, the StyleGAN model, and our mapping network. We also added sample images for such failure cases. (as suggested by R4 and as another analysis suggested by R3)
- **[New baselines]** **[R3]** We evaluated two other baselines for blackbox attacks (M2 and M4 in Table 4). The new results confirm that our method still outperforms other methods in the literature. (as another analysis suggested by R3)
- **[New sample reconstructed face images]** **[R3]** We added new sample reconstructed face images for different attacks against ArcFace and discussed the results. (as another analysis suggested by R3)
- **[More explanations for the proposed method and defined attcks]** **[R3]** We added further explanations for the proposed method and defined attacks (as suggested by R3). As also mentioned in the submitted manuscript, we will publish the source code of our paper, so that other researchers can reproduce our results and build upon our work. The source code will be complementary for any missing details.
- **[More discussions]** **[R2, R3, and R4]** We added further discussions to our experiments, including discussion on the results of our new ablation study (as suggested by R2), limitations and failure cases (as suggested by R4) and new baselines (for further analyses as suggested by R3).
- **[Mitigation of such attacks]** **[R1 and R4]** We revised the "Ethics Statement" section and added a new paragraph explaining how to mitigate such kinds of attacks. We added citations to some references for biometric template protection algorithms to provide readers with references to ways to mitigate such attacks. (as suggested by R1 and R4)

For simplicity, we use the following numbers to refer to each reviewer in our responses (either this general response or individual responses):
- R1: Reviewer e7hV
- R2: Reviewer 3DHW
- R3: Reviewer dNW2
- R4: Reviewer bw96

Finally, we would like to kindly invite all reviewers to further discussions should they still have any doubts or concerns.

---

### Decision · Program_Chairs · 2023-01-20

**Decision:**

Reject

**Justification For Why Not Higher Score:**

Despite the net positive evaluations by the reviewers, the relatively small and incremental contribution of the work makes it fall short of the bar for acceptance to ICLR. The positive opinions of the reviewers were mostly based on the application and the considered application scenarios (amply clarified during the reviewer-AC teleconference).

**Justification For Why Not Lower Score:**

N/A

**Metareview: Summary, Strengths And Weaknesses:**

# Summary of Contribution

This paper describes an approach to template-inversion attach against face recognition systems. The authors propose a generative model to reconstruct face images from facial templates. Using a pre-trained StyleGAN, whose encoder space is aligned with facial template images, and the StyleGAN encoder, the proposed system reconstructs face images that are able to fool trained facial recognition systems with high accuracy. The authors show through experimental results how the approach can be applied in blackbox and whitebox scenarios.

# Strengths

+ **Generality**: The main strength of the paper lies in its methodical consideration of a range of application attack scenarios. The authors define five different scenarios that depend on the knowledge of the face recognition system available to the attacker.

+ **Empirical Evaluation**: The experimental results provide convincing evidence that the proposed approach is effective in blackbox and whitebox scenarios.

# Weaknesses

+ **Novelty**: The novelty of the approach is quite limited as it is largely based on StyleGAN for face generation, with the addition of latent space alignment with facial template images (something which has become standard practice in the broader learning literature using GANs). The *application* is novel, but the overall approach is incremental with respect to the existing literature.

+ **Focus of Contribution**: Related to the previous point, the main contribution of this work is on the adversarial attack application and not so much on the representation learning. To be clear, the application is novel and the empirical evaluation is convincing, however given the focus of the contribution it seems somewhat out of place at ICLR.

+ **Limited Comparisons**: The four compared techniques from the literature (Table 2) are somewhat limited, in particular the more recent reported results of Duong et al. (2020) and Truong et al. (2022) are significantly better than the included baselines and are comparable (at least on LFW) to the proposed approach. While it is understandably limiting the lack of published code, a discussion of them (and even qualified inclusion in the table) along with a critical comparison with proposed approach is in order.

# Summary

There was a lively discussion among the reviewers and area chair about this paper. The reviewers and AC were all in agreement about the quality of the submitted paper, however they were equally unanimous in their opinion that the main contribution  of the work is primarily focused on the adversarial attack application scenarios and that the technical/theoretical contributions to representation learning are incremental and limited.

Finally, there is additional related literature that should be carefully considered, for example:

      Yang, Lu, Qing Song, and Yingqi Wu. "Attacks on state-of-the-art
      face recognition using attentional adversarial attack generative
      network." Multimedia tools and applications 80.1 (2021):
      855-875.

which uses an attentional VAE, and:

      Li, Dongze, et al. "Exploring adversarial fake images on face
      manifold." Proceedings of the IEEE/CVF Conference on Computer
      Vision and Pattern Recognition. 2021.

which takes StyleGAN latent-space approach similar to that proposed in this paper (although the specific goal is not adversarial attacks on FR systems).

In summary, while the paper makes the empirical case for the proposed approach, it does not contain enough novel contribution to meet the bar for acceptance at ICLR.


**Summary Of Ac-Reviewer Meeting:**

During the teleconference, which all four reviewers attended, all reviewers expressed an overall lack of enthusiasm for the submitted paper on the basis of its contributions to representation learning in general. There was unanimous agreement that the (i) the paper is well-written; (ii) that it addresses an interesting application; and (iii) that the empirical results are interesting and convincing. However there was equally unanimous agreement that the paper, given its primarily application-oriented contributions, was not a good fit for ICLR since the technical/theoretical contributions are incremental and minor. More than one reviewer commented that the paper would be a much better fit for a vision conference rather than a machine learning venue.